# Comparative Evidence on Corporate Governance Outcomes in the G20 Countries

**Voicu D. Dragomir** 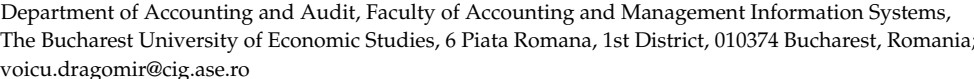

Department of Accounting and Audit, Faculty of Accounting and Management Information Systems, The Bucharest University of Economic Studies, 6 Piata Romana, 1st District, 010374 Bucharest, Romania; voicu.dragomir@cig.ase.ro

**Abstract:** The purpose of this study is to investigate the differences between developed countries in terms of corporate governance outcomes at aggregate and granular levels. The population of companies was collected from the database curated by Refinitiv. The sample was selected according to two criteria: the existence of governance scores for the financial year 2021 and the registration of a company in any of the G20 countries or the European Union. The results are presented by ranking the G20 countries based on four aggregate indicators and four granular indicators of corporate governance quality. While the differences regarding the aggregate indicators are not statistically strong, the intercountry differences on board independence, board gender diversity, board skills, and auditor tenure are especially relevant. The present article opens an avenue of research on international corporate governance linked to cultural dimensions, comparative legal systems, national approach to corporate social responsibility, and corporate governance principles.

**Keywords:** corporate governance; international comparison; G20; director independence; board gender diversity; CSR strategy; shareholder rights

## 1. Introduction

Corporate governance is the regulatory and procedural environment that affects decision-making processes at the firm level [1]. The regulatory environment includes the company law, financial market regulations, corporate governance codes, and company by-laws. On the other hand, corporate governance outcomes are the implementations, within each firm, of national or market-level corporate governance requirements [2]. Practical arrangements refer to the structure of the board of directors, the existence of board committees, skills of board members, executive compensation arrangements, shareholder rights, voting rules in the general meeting of shareholders, stakeholder engagement, and many other aspects [3]. At the international level, there is significant convergence between corporate governance requirements, although many path-dependent and contextual differences remain [4]. The research question of this paper is: How different are several developed countries in terms of their corporate governance outcomes, as of the year 2021?

International comparisons of corporate governance systems and outcomes have been conducted for the past three decades [5]. The best known paper in this area is the contribution of La Porta et al. [6]. The authors considered the legal protection of investors as the most important aspect of corporate governance. Strong investor protection is the prerequisite for effective corporate governance, as reflected in the existence of a strong financial market, dispersed ownership, and efficient capital allocation. Shareholder voting systems [7] are one of the main ways to enact "corporate democracy" [8], arguing for the advantage of board diversity in better decision-making. Recent articles argue that awarding the highest priority to shareholders or any specific group of stakeholders leads to an inefficient allocation of collective welfare [9].

The traditional classification of corporate governance models has been the Continental European model vs. the Anglo-Saxon model [10]. This is a very basic classification that

is not supported by evidence [11]. In contrast, a study [12] on corporate governance practices in BRICS countries (Brazil, Russia, India, China, and South Africa) and another study [13] on South Asian countries found that there are significant differences in corporate governance disclosures, and that institutional theory [14] is apt to investigate the normative and coercive pressures that shape national legal systems. On the other hand, agency theory is capable of explaining the antecedents and consequences of governance codes, which are, in many markets, the regulatory source of governance systems [15]. This is especially applicable to multinational enterprises [16].

International corporate governance [17] is an area of research that seeks to explore factors of convergence and divergence in corporate governance systems and outcomes in several societies and economies [18]. The rule of law and shareholder protection mechanisms are particularly important in supporting the corporate governance system. Alongside wider institutional systems and pressures, governance arrangements can influence the internationalization of companies and their resilience in turbulent markets [19]. A recent study on 23 countries [20] reported that firms in common law countries have better financial performance when the legal systems provide greater investor protection. The reverse is true in the sense that weaker investor protection in civil law countries is associated with a negative valuation.

A comparative analysis of corporate governance systems and outcomes cannot rely on a single classification method [11]. National cultural practices have a strong influence on the institutional environment, the main factor of corporate governance arrangements [21]. Corporate governance outcomes are understood as measures of board accountability and diversity, internal controls, audit quality, executive compensation, ownership configurations, shareholder rights, and stakeholder engagement. The present paper relies on a comprehensive assessment of a country's corporate governance system by taking into account several constructs from the literature. These corporate governance outcomes are different from country to country, and are not likely to become completely harmonized in the near future [22].

The contribution of the present research is to investigate the differences in corporate governance arrangements in different countries, using a large sample of companies listed on international stock exchanges. Firms incorporated in the G20 countries were considered a relevant sample because the G20 includes the G7 group of developed countries (Canada, France, Germany, Italy, Japan, the United Kingdom and the United States), the European Union (represented by the President of the European Commission), and the most important BRICS countries [23]. This group of countries accounts for 80% of global GDP [24] so that there is a degree of compatibility between the corporate governance systems of these countries. The present research will assess the degree of similarity using the corporate governance scores of sample companies, calculated by the financial agency Refinitiv on a homogenous scale.

The paper is organized as follows. The main source of data is described in relation to the corporate governance variables of interest. The sample is described for the G20 countries. The applied method of analysis is one-way ANCOVA, with company size as a covariate. Country rankings are presented according to several criteria: the overall corporate governance score, shareholder rights score, management quality score, corporate social responsibility (CSR) score, board independence, board gender diversity, board-specific skills, and auditor tenure. Conclusions offer more insight into the research domain of international corporate governance.

## 2. Materials and Methods

### 2.1. Data Source

The proprietary database curated by the financial agency Refinitiv [25] has been used in scientific research on the topic of corporate governance [26–28]. This database is a comprehensive and trusted source for investment purposes. For the largest listed companies, Refinitiv calculates a corporate governance score [29], which represents the

relative sum of category weights that are uniform between industries, for multiple criteria related to management, shareholder rights, and CSR strategy. In total, Refinitiv collects 138 corporate governance indicators for each company analyzed in the October 2022 edition.

The management score includes data (70 indicators) related to corporate boards (size, functions, structure, attendance, independence, skills, and diversity), compensation (policy, targets, incentives, restrictions, and committees), CEO–Chairman separation, the succession plan, internal audit, and audit committee independence. The shareholder rights dimension evaluates data (39 indicators) on equal shareholder rights and specific policies, shareholders voting on executive pay, director election requirements, veto power, state-owned companies, non-audit to audit fees ratio, and auditor tenure. The CSR (corporate social responsibility) strategy dimension (29 indicators), which Refinitiv considers to be part of the corporate governance pillar, refers to the existence of a sustainability committee, stakeholder engagement, sustainability reporting, and external assurance of nonfinancial reports.

### 2.2. Sample Selection

The data were collected from the *Industries* catalogue of companies, as compiled by Refinitiv. The population of companies is divided by industry as follows: Energy (2360), Basic Materials (7856), Industrials (9878), Consumer Cyclicals (8842), Consumer Non-Cyclicals (4423), Financials (9003), Healthcare (5033), Technology (8084), Utilities (1350), Real Estate (3597), Associations (3), Government Activity (7), and Academic and Educational Services (288). All these organizations are classified as for-profit.

The first selection criterion was the availability of the Corporate Governance Score for the financial year 2021. The resulting sample had 8408 observations. The second selection criterion was for the country of incorporation (i.e., "country of domicile" in Refinitiv) to be part of the G20. Considering that the European Union is a member of the G20, companies from the 27 member states were considered eligible for analysis. The sample comprises 6288 observations for 19 countries in the G20 and 1136 observations for the European Union (but only companies in 21 member states had data collected by Refinitiv). These two subsets of data overlap for Germany, France, and Italy, which are members of the European Union and the G20. The sample is cross-sectional for the financial year 2021, as it reflects the latest version of the Refinitiv database (as of 1 October 2022). All companies included in the sample are very large corporations, listed on global stock exchanges. A summary of the sample selection procedure is presented in Figure 1.

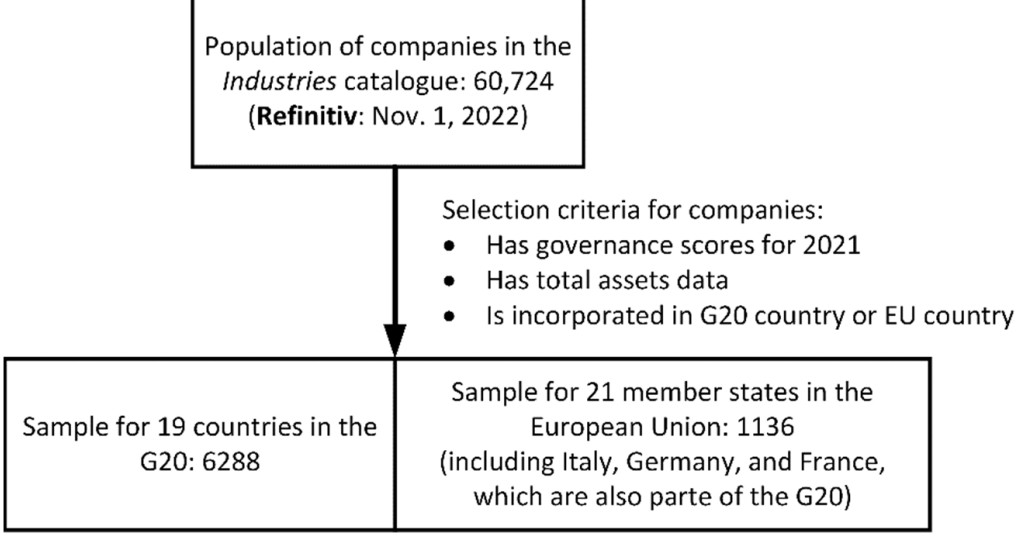

**Figure 1.** A summary of sample selection.

*2.3. Hypothesis and Variables*

The tested hypothesis is as follows:

**Hypothesis 1:** *Corporate governance outcomes are significantly different between the G20 countries, after adjusting for company size.*

The corporate governance outcomes are extracted from the Refinitiv database and are listed below:

- The Governance Pillar Score (*GovScore*) is a weighted score on a scale of 0 to 100 and measures the company's systems and processes, ensuring that board members and executives act in the interest of shareholders [25]. It reflects the creation of managerial incentives and the system of checks and balances to generate long-term shareholder value. This score is normalized by industry and its calculation algorithm is not disclosed by Refinitiv. The Governance Pillar Score is a weighted average of the Management Score, the Shareholder Rights Score, and the CSR Strategy Score. The Governance Pillar Score has been used in the literature as a proxy of corporate governance quality [29].
- Management Score (*ManScore*) is a weighted score on a scale of 0 to 100 and measures a company's commitment and effectiveness towards following best practice governance practices [25]. Some relevant indicators are the existence of policies related to the audit committee, nomination committee, remuneration committee; board structure, independence, diversity, skills, and tenure; audit committee structure, independence, and expertise; remuneration committee independence; executive compensation policies and targets; ethnic minorities as board members. This score is normalized by industry and its calculation algorithm is not disclosed by Refinitiv.
- Shareholder Rights score (*ShareScore*) is a weighted score on a scale of 0 to 100 and measures the company's effectiveness towards the equal treatment of shareholders and the use of anti-takeover devices [25]. Some relevant indicators are policies related to shareholder rights, equal voting rights, different voting rights, director election majority, and the advance notice period; limitation of director liability; litigation expenses; auditor tenure. This score is normalized by industry and its calculation algorithm is not disclosed by Refinitiv.
- The CSR Strategy Score (*CsrScore*) is a weighted score on a scale of 0 to 100 and reflects the company's efforts to integrate the economic, social and environmental dimensions of the business model into the decision-making process [25]. In the literature, this score has been used as a proxy for the company's capacity of integrated thinking [30,31]. It refers to the following aspects: the existence of a sustainability committee; the integration of financial and non-financial factors in the management report; compliance with the Global Reporting Initiative; stakeholder engagement; reporting on the Sustainable Development Goals. Thus, CSR can be understood as a function of the corporate governance system [32]. This score is normalized by industry and its calculation algorithm is not disclosed by Refinitiv.
- Board independence (*BoardIndep*) is the percentage of independent board members reported by the company. From the perspective of agency theory, a higher proportion of board independence is widely considered as one of the most important aspects of good corporate governance [33,34]. From the perspective of stakeholder theory, board credibility and company reputation are enhanced by the presence of independent directors [35]. From the perspective of signaling theory, board independence is a positive signal to the market that leads to reduced information asymmetry and ensures favorable responses from different stakeholders [36].
- Board gender diversity (*GenDiv*) is the percentage of female members on the board of directors. In many countries around the world [37], a higher proportion of gender diversity is considered best governance practice [38] because it supports the interests of shareholders [39] and other stakeholders.

- Board-specific skills (*BoardSkills*), measured as the percentage of board members who have either an industry-specific background or a strong financial background [40]. From the perspective of agency theory, board-specific skills can support the monitoring function of the board [41,42]. This aspect of good governance has scarcely been explored in the literature, probably due to a lack of reliable data at the international level.
- Auditor Tenure (*AuditTenure*) is the number of years the current auditor is providing services to the organization. A longer audit tenure is considered to adversely affect the quality of audit reports, and, by consequence, the quality of the corporate governance system [43]. However, other authors provided evidence that increased auditor tenure reduces the probability of earnings management [44]. Refinitiv assigns a positive valence to a longer auditor tenure.

### 2.4. Statistical Procedures

The relevant statistical procedure is ANCOVA (analysis of covariance). This procedure is a more complex version of ANOVA (the analysis of variance) because it also introduces a covariate. In this study, the covariate is company size, proxied by the natural logarithm of total assets converted to the same currency. Larger companies are expected to be more visible and face pressure to adopt better governance practices. Moreover, company size is a standard covariate/control variable in governance-related studies [45].

Regarding the assumption of the ANCOVA procedure, the following tests are discussed [46]. There is an interaction between the categorical dependent variable (a specific country) and the covariate (the average size of companies registered in that country). For example, the largest companies in the world are in the G7 countries (not G20), so that the homogeneity of regression slopes is not applicable. Given that the group sizes are very different, homogeneity of variance is not expected [47,48]. Finally, scatter plots of governance outcomes vs. the covariate (*lnTA*) were inspected and nonlinear patterns were not identified.

To calculate the F-value of the grouping variable, the heteroscedasticity correction was applied to the coefficient of covariance matrix. The post-hoc tests apply the Bonferroni correction. Significant differences between groups are reported at the $p < 0.01$ threshold, for a more conservative assessment. A fragment of software code written in R is presented as Algorithm A1 in Appendix A, indicating the statistical procedures for ANCOVA and the post-hoc tests [49].

As explained by Field [46], ANCOVA is specifically designed to test hypotheses regarding group differences, while eliminating the effect of confounding variables. It is similar to multiple regression, but ANCOVA benefits from the application of post-hoc tests between groups. Multiple regression only compares the selected groups (using dummy variables) to a base category, while post-hoc tests in ANCOVA compare all pair of groups and identify significant differences while ensuring that the cumulative Type I error is below 0.05. Moreover, the present research design cannot set a "base" group (i.e., a country of reference) as required by multiple regression, because that would not make sense. Therefore, ANCOVA is the appropriate procedure to answer the research question.

## 3. Results

### 3.1. Hypothesis Testing for the Governance Pillar Score

The results in Table 1 show the adjusted mean scores for the Governance Pillar score (*GovScore*) by country/group of countries. The F-statistic of the grouping variable is $F (19, 7403) = 20.46$ ($p < 0.001$), after controlling for company size. The highest average score belongs to South Korea, but the sample is relatively small. Only South Korea and Germany have an average score above 60. At the opposite end of the scale, China and Japan have an average *GovScore* of around 45, significantly lower than the United States and the European Union. Therefore, the main hypothesis of the study is confirmed for the variable *GovScore*. The results do not prove the superiority of the common law system.

**Table 1.** The analysis of covariance of *GovScore* by country/group of countries, adjusted for company size, in descending order.

| G20 Member | Sample | *GovScore* Adj. Mean | SE | Sig. Differences [1] |
|---|---|---|---|---|
| South Korea (KR) | 22 | 65.1 | 4.21 | JP, CN |
| Germany (DE) | 186 | 63.9 | 1.45 | IN, JP, US, CN |
| United Kingdom (GB) * | 527 | 57.4 | 0.86 | JP, US, CN |
| Italy (IT) | 82 | 57.2 | 2.18 | JP, CN |
| European Union (EU) | 1136 | 56.8 | 0.58 | JP, US, CN, DE |
| Turkey (TR) | 73 | 56.4 | 2.31 | JP, CN |
| Indonesia (ID) | 39 | 55.8 | 3.16 | - |
| Australia (AU) * | 358 | 55.7 | 1.05 | CN, JP, DE |
| Brazil (BR) | 36 | 54.6 | 3.29 | - |
| France (FR) | 148 | 54.2 | 1.62 | CN, DE |
| United States (US) * | 2540 | 52.0 | 0.39 | JP, GB, EU, DE |
| Canada (CA) * | 295 | 51.6 | 1.15 | DE, JP, CN |
| Mexico (MX) | 69 | 51.1 | 2.37 | DE |
| South Africa (ZA) | 114 | 51.1 | 1.85 | DE |
| India (IN) * | 221 | 50.7 | 1.33 | DE, GB, EU |
| Russia (RU) | 16 | 50.3 | 4.93 | - |
| Saudi Arabia (SA) ** | 28 | 49.5 | 3.73 | - |
| Argentina (AR) | 24 | 49.2 | 4.02 | - |
| China (CN) | 1105 | 46.3 | 0.60 | DE, EU, GB, US, AU |
| Japan (JP) | 405 | 44.5 | 0.99 | US, IT, GB, FR, EU |

[1] Significant difference at $p < 0.01$. * Common law system, ** Islamic law. The rest have a civil law system or mixed types.

### 3.2. Hypothesis Testing for the Management Score

The results in Table 2 show the adjusted mean scores for the Management Score (*ManScore*) by country/group of countries. The F-statistic of the grouping variable is $F_{(19, 7403)} = 14.72$ ($p < 0.001$), after controlling for company size. The highest average score belongs to South Korea, but its sample is relatively small. Only South Korea and Germany have an average score above 60. At the opposite end of the scale, Argentina, China, and Japan have an average *ManScore* below 50, significantly lower than the United States and the European Union. Therefore, the main hypothesis of the study is confirmed for the variable *ManScore*. The results do not prove the superiority of the common law system.

### 3.3. Hypothesis Testing for the Shareholder Rights Score

The results in Table 3 show the adjusted mean scores for the Shareholder Rights Score (*ShareScore*) by country/group of countries. The F-statistic of the grouping variable is $F_{(19, 7403)} = 7.57$ ($p < 0.001$), after controlling for company size. The highest average score belongs to Germany, followed by South Korea. Only South Korea and Germany have average scores above 60. At the opposite end of the scale, Canada, China, and Japan have an average *ShareScore* below 50, significantly lower than the Germany. Therefore, the hypothesis of the study is confirmed for the variable *ShareScore*, but only between the extremes of the scale. All other countries are not significantly different from each other in terms of shareholder protection. The results do not prove the superiority of the common law system. Surprisingly, shareholder protection appears to be the strongest in Germany, a continental European country known for its stakeholder activism [50] and a more inclusive model of corporate governance [51].

**Table 2.** The analysis of covariance of *ManScore* by country/group of countries, adjusted for company size, in descending order.

| G20 Member | Sample | *ManScore* Adj. Mean | SE | Sig. Differences [1] |
|---|---|---|---|---|
| South Korea (KR) | 22 | 74.0 | 5.45 | JP, CN |
| Germany (DE) | 186 | 66.8 | 1.87 | US, EU |
| Italy (IT) | 82 | 59.7 | 2.82 | JP, CN |
| European Union (EU) | 1136 | 58.5 | 0.75 | CN |
| United Kingdom (GB) * | 527 | 58.2 | 1.12 | CN |
| Brazil (BR) | 36 | 58.1 | 4.26 | - |
| Indonesia (ID) | 39 | 58.0 | 4.09 | - |
| Australia (AU) * | 358 | 57.5 | 1.36 | JP, CN |
| Turkey (TR) | 73 | 56.6 | 2.99 | - |
| United States (US) * | 2540 | 56.1 | 0.51 | CN, DE |
| France (FR) | 148 | 55.7 | 2.10 | - |
| Saudi Arabia (SA) ** | 28 | 54.2 | 4.83 | - |
| Canada (CA) * | 295 | 54.1 | 1.49 | DE |
| Mexico (MX) | 69 | 53.3 | 3.08 | - |
| South Africa (ZA) | 114 | 51.4 | 2.39 | DE |
| India (IN) * | 221 | 51.3 | 1.72 | DE |
| Russia (RU) | 16 | 50.0 | 6.39 | - |
| Argentina (AR) | 24 | 49.6 | 5.22 | - |
| China (CN) | 1105 | 47.4 | 0.77 | US, GB, EU, DE |
| Japan (JP) | 405 | 46.2 | 1.29 | US, GB, EU, DE |

[1] Significant difference at $p < 0.01$. * Common law system, ** Islamic law. The rest have a civil law system or mixed types.

**Table 3.** The analysis of covariance of *ShareScore* by country/group of countries, adjusted for company size, in descending order.

| G20 Member | Sample | *ShareScore* Adj. Mean | SE | Sig. Differences [1] |
|---|---|---|---|---|
| Germany (DE) | 186 | 64.8 | 1.92 | US, JP, IN |
| South Korea (KR) | 22 | 60.1 | 5.59 | - |
| Russia (RU) | 16 | 59.4 | 6.56 | - |
| European Union (EU) | 1136 | 57.1 | 0.77 | JP |
| Turkey (TR) | 73 | 56.9 | 3.07 | - |
| France (FR) | 148 | 56.7 | 2.16 | - |
| United Kingdom (GB) * | 527 | 56.7 | 1.14 | JP |
| Indonesia (ID) | 39 | 55.7 | 4.20 | - |
| Italy (IT) | 82 | 54.8 | 2.90 | - |
| United States (US) * | 2540 | 54.6 | 0.52 | JP |
| Australia (AU) * | 358 | 54.5 | 1.40 | DE |
| Mexico (MX) | 69 | 54.1 | 3.16 | - |
| Saudi Arabia (SA) ** | 28 | 53.6 | 4.96 | - |
| Brazil (BR) | 36 | 53.2 | 4.37 | - |
| India (IN) * | 221 | 51.2 | 1.76 | DE |
| South Africa (ZA) | 114 | 50.8 | 2.46 | DE |
| Argentina (AR) | 24 | 50.7 | 5.35 | - |
| Canada (CA) * | 295 | 49.7 | 1.53 | DE, EU |
| China (CN) | 1105 | 48.8 | 0.79 | US, GB, EU, DE |
| Japan (JP) | 405 | 48.5 | 1.32 | DE, EU, GB, US |

[1] Significant difference at $p < 0.01$. * Common law system, ** Islamic law. The rest have a civil law system or mixed types.

### 3.4. Hypothesis Testing for the CSR Strategy Score

The results in Table 4 show the adjusted mean scores for the CSR Strategy Score (*CsrScore*) by country/group of countries. The F-statistic of the grouping variable is $F (19, 7403) = 52.63$ ($p < 0.001$), after controlling for company size. The highest aver-

age score belongs to Turkey (for a relatively small sample), closely followed by the United Kingdom. At the opposite end of the scale, Japan, South Korea, the United States, and Saudi Arabia have an average *CsrScore* below 30. There are numerous significant differences between the average scores of the G20 members. The range of average scores is much larger for *CsrScore* than for the previous dimensions of corporate governance. In comparison to shareholder rights, the CSR/sustainability domain is less regulated. The high average score of Turkish companies cannot be clearly explained by previous literature [52]. However, integrated reports are mandatory in South Africa [53], and non-financial reports are mandatory in the European Union [54,55]. The London Stock Exchange requires CSR reporting (which includes only environmental and social information) for all firms listed on its main market [56], which explains the high average score for the UK. In conclusion, the main hypothesis of the study is confirmed for the variable *CsrScore*.

**Table 4.** The analysis of covariance of *CsrScore* by country/group of countries, adjusted for company size, in descending order.

| G20 Member | Sample | *CsrScore* Adj. Mean | SE | Sig. Differences [1] |
|---|---|---|---|---|
| Turkey (TR) | 73 | 55.0 | 3.14 | US, SA, CN |
| United Kingdom (GB) * | 527 | 54.2 | 1.17 | US, SA, MX, JP |
| South Africa (ZA) | 114 | 49.8 | 2.51 | US, SA, CN |
| Australia (AU) * | 358 | 48.5 | 1.43 | US, SA, JP, CN |
| Italy (IT) | 82 | 48.3 | 2.96 | US, JP, SA |
| Germany (DE) | 186 | 47.6 | 1.96 | US, SA, JP |
| European Union (EU) | 1136 | 47.1 | 0.79 | US, SA, JP, GB |
| India (IN) * | 221 | 46.3 | 1.80 | US, JP, SA, CN |
| Indonesia (ID) | 39 | 45.0 | 4.29 | US |
| Argentina (AR) | 24 | 44.8 | 5.47 | - |
| France (FR) | 148 | 43.0 | 2.21 | US, JP, SA, GB |
| Canada (CA) * | 295 | 41.9 | 1.56 | US, JP, GB |
| Brazil (BR) | 36 | 40.7 | 4.47 | - |
| Russia (RU) | 16 | 37.3 | 6.70 | - |
| China (CN) | 1105 | 36.6 | 0.81 | US, GB, EU, DE |
| Mexico (MX) | 69 | 35.2 | 3.23 | TR, GB |
| Japan (JP) | 405 | 29.7 | 1.35 | ZA, TR, CN |
| South Korea (KR) | 22 | 28.4 | 5.72 | TR, GB |
| United States (US) * | 2540 | 27.2 | 0.53 | TR, GB, EU, CN |
| Saudi Arabia (SA) ** | 28 | 19.7 | 5.07 | US, CA, GB, EU |

[1] Significant difference at $p < 0.01$. * Common law system, ** Islamic law. The rest have a civil law system or mixed types.

### 3.5. Hypothesis Testing for Board Independence

The results in Table 5 show the adjusted mean values for board independence (*BoardIndep*) by country/group of countries. The F-statistic of the grouping variable is $F_{(19, 7386)} = 684.55$ ($p < 0.0001$), after controlling for company size. Data for this indicator are unavailable for 17 companies, which have been eliminated from the analysis. The United States and Canada have the highest average proportion of independent board members, above 70%. Australia, South Africa, and the United Kingdom have an average value greater than 60%. At the opposite end of the scale, only Argentina (with a sample of 24 companies) has an average proportion of board independence below 30%. The differences between countries are highly significant, thus confirming the hypothesis of the study. The results also indicate that companies in common law countries aim for a higher proportion of independent board members. This result has previously been confirmed in the literature [57].

**Table 5.** The analysis of covariance of *BoardIndep* by country/group of countries, adjusted for company size, in descending order.

| G20 Member | Sample | *BoardIndep* (%) Adj. Mean | SE | Sig. Differences [1] |
|---|---|---|---|---|
| United States (US) * | 2539 | 79.6 | 0.34 | GB, FR, EU, DE, CN |
| Canada (CA) * | 295 | 76.7 | 1.02 | EU, CN, FR, GB, IN |
| Australia (AU) * | 355 | 67.2 | 0.93 | CA, CN, DE, EU, FR |
| South Africa (ZA) | 114 | 64.6 | 1.63 | US, CN, AR, TR |
| United Kingdom (GB) * | 526 | 63.5 | 0.76 | IN, ID, JP, MX |
| European Union (EU) | 1134 | 58.7 | 0.52 | FR, GB, IN, JP |
| South Korea (KR) | 22 | 54.9 | 3.72 | US, TR, CA |
| Italy (IT) | 82 | 53.8 | 1.92 | US, JP, TR, CN, AU |
| Mexico (MX) | 69 | 50.8 | 2.10 | US, TR, ZA, CA |
| India (IN) * | 220 | 47.9 | 1.17 | US, JP, TR, AU |
| Brazil (BR) | 36 | 47.4 | 2.90 | CA, GB, US, ZA, AU |
| Indonesia (ID) | 39 | 47.4 | 2.79 | US, ZA, CA, AU |
| Russia (RU) | 16 | 46.6 | 4.36 | US, AR, AU |
| France (FR) | 148 | 46.4 | 1.43 | GB, JP, TR |
| Germany (DE) | 186 | 43.2 | 1.28 | EU, GB, ZA, CA |
| Saudi Arabia (SA) ** | 28 | 42.4 | 3.29 | US, ZA, AU |
| China (CN) | 1100 | 38.3 | 0.53 | FR, GB, IN, MX, ZA |
| Japan (JP) | 402 | 35.6 | 0.88 | US, MX, ZA, KR, AU |
| Turkey (TR) | 72 | 32.4 | 2.05 | US, ZA, GB, EU, AU |
| Argentina (AR) | 24 | 21.2 | 3.56 | US, CA, AU, EU, etc. |

[1] Significant difference at $p < 0.01$. * Common law system, ** Islamic law. The rest have a civil law system or mixed types.

### 3.6. Hypothesis Testing for Gender Diversity

The results in Table 6 show the adjusted mean values of board gender diversity (*GenDiv*) by country/group of countries in the G20. The F-statistic of the grouping variable is $F_{(19, 7388)} = 247.62$ ($p < 0.0001$), after controlling for company size. France has an average proportion of more than 40% women on boards, a result significantly different from countries with 30% or less. At the opposite end of the spectrum, Japan has less than 10% female board members, on average, in a sizable sample of 402 companies. These results match the study [58] by N26 on female opportunity around the world in 2021. Nordic countries, UK, Germany, France, Baltic countries, and New Zealand (not in the G20) occupy the first ten positions of this ranking. Japan is at position 52 of 100, while Saudi Arabia occupies the 93rd place. Therefore, the hypothesis of the current study is validated regarding country differences based on board gender diversity. Research has shown that female directorship acts as a catalyst and determines norm changes on the board, leading to improved governance [59]. In addition, independent female directors are effective at changing board processes and improving governance outputs, such as the company's environmental policy. However, national gender inequality negatively moderates this relationship [60].

### 3.7. Hypothesis Testing for Board Skills

Director skills may be any of the following [61]: company business, entrepreneurial, finance, governance, policy, leadership, legal, academic, management, manufacturing, marketing, risk management, technology, and sustainability. From this list, Refinitiv retained manufacturing and technology education linked to the respective industry, and general economic, finance and leadership education [25]. Research has shown that that boards whose directors have more commonality in skill sets have better firm performance [61].

**Table 6.** The analysis of covariance of *GenDiv* by country/group of countries, adjusted for company size, in descending order.

| G20 Member | Sample | *GenDiv* (%) Adj. Mean | SE | Sig. Differences [1] |
|---|---|---|---|---|
| France (FR) | 148 | 43.4 | 1.01 | US, GB, JP, KR, etc. |
| Italy (IT) | 82 | 37.4 | 1.35 | GB, DE, CN, CA, etc. |
| European Union (EU) | 1134 | 32.6 | 0.36 | FR, US, JP, KR, etc. |
| South Africa (ZA) | 114 | 32.2 | 1.15 | US, DE, CA, CN, etc. |
| United Kingdom (GB) * | 526 | 30.1 | 0.53 | US, TR, SA, RU, MX |
| Australia (AU) * | 355 | 27.8 | 0.65 | EU, CN, FR, IT, etc. |
| Canada (CA) * | 295 | 26.6 | 0.71 | EU, CN, FR, IT, etc. |
| Germany (DE) | 186 | 26.5 | 0.89 | FR, IN, IT, JP, etc. |
| United States (US) * | 2539 | 25.6 | 0.24 | CN, EU, FR, GB, etc. |
| Turkey (TR) | 72 | 17.3 | 1.44 | EU, DE, GB, FR, etc. |
| India (IN) * | 220 | 16.8 | 0.82 | US, ZA, JP, SA, etc. |
| Brazil (BR) | 36 | 14.7 | 2.04 | EU, DE, FR, GB, etc. |
| China (CN) | 1102 | 13.6 | 0.37 | US, SA, JP, GB, FR |
| Russia (RU) | 16 | 13.2 | 3.06 | ZA, IT, EU, etc. |
| Indonesia (ID) | 39 | 11.0 | 1.96 | ZA, US, IT, GB, etc. |
| Argentina (AR) | 24 | 10.6 | 2.50 | EU, DE, FR, GB, etc. |
| South Korea (KR) | 22 | 10.0 | 2.61 | US, ZA, IT, DE, etc. |
| Mexico (MX) | 69 | 9.89 | 1.47 | US, ZA, IT, EU, etc. |
| Japan (JP) | 402 | 7.82 | 0.62 | US, ZA, IT, EU, etc. |
| Saudi Arabia (SA) ** | 28 | 1.34 | 2.31 | US, ZA, TR, etc. |

[1] Significant difference at $p < 0.01$. * Common law system, ** Islamic law. The rest have a civil law system or mixed types.

The results in Table 7 show the adjusted mean values for the proportion of directors with specialized skills (*BoardSkills*) by country/group of countries. The F-statistic of the grouping variable is $F_{(19, 7387)} = 202.19$ ($p < 0.0001$), after controlling for company size. For some countries, the results in this category are almost inverse to the gender diversity indicator. Japan has the highest proportion of skilled directors, closely followed by South Africa, the United States, the UK, and Canada, around 60%. In contrast, Germany has the lowest proportion of directors (less than 20%) with an industry background or financial expertise. Differences between countries are significant across the entire spectrum. Companies headquartered in the European Union have, on average, half the proportion of Japan, at about 30%. The hypothesis of the present study is confirmed in relation to the proportion of skilled board members in companies of the G20. Previous research has shown that corporate governance attributes such as board experience, the background and skills of board members are significantly related to the capital-asset ratio in the case of large banks [40]. The geographical distribution of these characteristics deserves further investigation.

*3.8. Hypothesis Testing for Auditor Tenure*

The results in Table 8 show the adjusted mean values for auditor tenure in years (*AuditTenure*) by country/group of countries. The F-statistic of the grouping variable is $F_{(19, 7385)} = 110.55$ ($p < 0.0001$), after controlling for company size. Corporations in Canada and the United States have average auditor tenures of more than 11 years. In contrast, companies in Turkey, South Korea, Saudi Arabia, India, and Brazil resort to auditor rotation after less than four years, on average. Mandatory auditor rotation can improve audit quality, but with increased costs [62]. Conversely, requiring auditors to keep a skeptical assessment of the client's financial situation may be a cost-effective solution without sacrificing audit quality [63]. In conclusion, the hypothesis of the study is confirmed for companies headquartered in the G20 with respect to auditor tenure.

**Table 7.** The analysis of covariance of *BoardSkills* by country/group of countries, adjusted for company size, in descending order.

| G20 Member | Sample | *BoardSkills* (%) Adj. Mean | SE | Sig. Differences [1] |
|---|---|---|---|---|
| Japan (JP) | 402 | 63.0 | 0.90 | KR, MX, TR, RU, SA |
| South Africa (ZA) | 114 | 61.1 | 1.67 | EU, DE, BR, RU |
| United States (US) * | 2539 | 58.6 | 0.35 | DE, CN, EU, etc. |
| United Kingdom (GB) * | 526 | 57.5 | 0.77 | ID, IN, IT, KR, MX |
| Canada (CA) * | 295 | 56.9 | 1.04 | DE, EU, FR, IN, AR |
| China (CN) | 1101 | 53.6 | 0.54 | FR, ID, JP, MX, SA, TR |
| Australia (AU) * | 355 | 48.3 | 0.95 | BR, CA, DE, EU, FR, etc. |
| India (IN) * | 220 | 47.7 | 1.20 | JP, MX, SA, US, ZA |
| Italy (IT) | 82 | 39.8 | 1.97 | US, ZA, JP, CN, CA |
| Russia (RU) | 16 | 39.3 | 4.46 | ZA, US, DE |
| Turkey (TR) | 72 | 39.1 | 2.10 | US, ZA, DE, CA |
| South Korea (KR) | 22 | 36.7 | 3.80 | US, ZA, CA, CN |
| Indonesia (ID) | 39 | 31.7 | 2.85 | IN, JP, US, ZA, CA |
| France (FR) | 148 | 31.0 | 1.47 | GB, IN, JP, US, ZA |
| Mexico (MX) | 69 | 30.9 | 2.14 | US, ZA, CA |
| Brazil (BR) | 36 | 30.6 | 2.97 | CA, CN, GB, IN, JP, US |
| European Union (EU) | 1134 | 29.8 | 0.52 | US, GB, IN, JP, CN, TR |
| Argentina (AR) | 24 | 23.0 | 3.64 | AU, CN, GB, US, IN, etc. |
| Saudi Arabia (SA) ** | 28 | 22.5 | 3.37 | US, ZA, GB, CA |
| Germany (DE) | 186 | 18.7 | 1.31 | EU, FR, GB, IN, IT, JP |

[1] Significant difference at $p < 0.01$. * Common law system, ** Islamic law. The rest have a civil law system or mixed types.

**Table 8.** The analysis of covariance of *AuditTenure* by country/group of countries, adjusted for company size, in descending order.

| G20 Member | Sample | *AuditTenure* (Years) Adj. Mean | SE | Sig. Differences [1] |
|---|---|---|---|---|
| Canada (CA) * | 294 | 11.9 | 0.33 | CN, DE, EU, FR, GB, etc. |
| United States (US) * | 2539 | 11.6 | 0.11 | ZA |
| Argentina (AR) | 24 | 8.58 | 1.17 | - |
| Japan (JP) | 402 | 8.42 | 0.29 | TR, US |
| Australia (AU) * | 358 | 7.62 | 0.31 | IT, DE, CA, CN, US |
| Mexico (MX) | 65 | 7.40 | 0.71 | US |
| South Africa (ZA) | 114 | 6.64 | 0.54 | US, CA, IN |
| United Kingdom (GB) * | 527 | 6.41 | 0.25 | IN, JP, US |
| European Union (EU) | 1132 | 5.84 | 0.17 | AU, IN, JP, US |
| France (FR) | 147 | 5.56 | 0.47 | JP, US |
| China (CN) | 1104 | 5.33 | 0.17 | JP, US, CA |
| Indonesia (ID) | 39 | 5.15 | 0.91 | US |
| Germany (DE) | 186 | 5.13 | 0.42 | JP, US |
| Italy (IT) | 80 | 4.49 | 0.64 | JP, US |
| Russia (RU) | 16 | 4.31 | 1.43 | US |
| Turkey (TR) | 73 | 3.88 | 0.67 | US, AU |
| South Korea (KR) | 21 | 3.79 | 1.25 | US |
| Saudi Arabia (SA) ** | 28 | 3.70 | 1.08 | US |
| India (IN) * | 221 | 3.67 | 0.38 | JP, US, ZA, MX |
| Brazil (BR) | 36 | 3.63 | 0.95 | JP, CA, US |

[1] Significant difference at $p < 0.01$. * Common law system, ** Islamic law. The rest have a civil law system or mixed types.

## 4. Discussion

Country-level legal and regulatory institutions influence foreign ownership, foreign directorship, access to external financial capital, and cross-border M&A activity [64]. While international differences on regulatory issues in corporate governance have been thoroughly addressed in the past [6] and in similar research, the practical outcomes of corporate governance have not been subject to sufficient scrutiny. The present article uses the comprehensive database of Refinitiv to analyze and compare the aggregate scores of corporate governance outcomes, at the international level. The study draws eight indicators from the Refinitiv database and applies analysis of covariance (ANCOVA) to measure the differences

between countries on average corporate governance outcomes. The sample is composed of the members of G20, i.e., 19 countries and the European Union. The results are summarized in Table 9.

**Table 9.** A summary of results.

| Variable | Hypothesis Status | Sig. Differences |
|---|---|---|
| Governance Pillar Score | Confirmed | Between the upper half and lower half of the sample |
| Management Score | Confirmed | Between best performers and worst performers |
| Shareholder Rights Score | Confirmed | Between best performers and worst performers |
| CSR Score | Confirmed | Between the upper half and lower half of the sample |
| Board independence | Confirmed | Between almost all countries in the sample |
| Board gender diversity | Confirmed | Between almost all countries in the sample |
| Board-specific skills | Confirmed | Between almost all countries in the sample |
| Auditor tenure | Confirmed | Between the upper half and lower half of the sample |

The results have a double valence. First, the aggregate indicators (*GovScore*, *ManScore*, *ShareScore*, and *CsrScore*) have country-level mean values that are significantly different only between the best performers and the worst performers. The countries with average rankings are not significantly different from each other. Second, the granular indicators (*BoardIndep*, *GenDiv*, *BoardSkills*, and *AuditTenure*) are significantly different between most countries in the sample. For these indicators, the range is larger, and the standard errors are smaller. Therefore, there may be significant differences between the best performer and the average performer in each category. Moreover, the surprising result is that a country can be a best performer in one category and a near-worst performer in another category. This would explain why aggregate indicators such as *GovScore* are not sensitive to real differences in corporate governance outcomes.

Country-level differences become more visible on the granular dimensions of corporate governance. For example, the common law system (present in the US, the UK, Canada, Australia) is a significant influence on board independence. European countries have a mandate to reach a higher proportion of female directors, as agreed by the European Parliament and member country negotiators [65]. Surprisingly, companies headquartered in continental European countries are not keen on appointing directors with industry skills, Germany being the worst performer in this category. Finally, auditor tenure is significantly higher in North American countries (the US and Canada) than in any other country of the G20. This is surprising, considering the past scandals involving financial auditors in the US [66].

## 5. Conclusions

The present study is the first contribution that uses a large international sample to investigate intercountry differences in corporate governance outcomes. The insights on granular indicators of governance outcomes can be further linked to Hofstede's cultural dimensions [67]. This is a major avenue of research that can successfully complement the present approach of calculating country aggregate indicators. Furthermore, the influence of the legal system on corporate governance outcomes is an ongoing debate [64]. Granular indicators of governance practices can be further explained by the configuration of legal systems, at national or supra-national level (like in the European Union). Finally, the corporate social responsibility dimension merits further research. In the present sample, this aggregate dimension had the largest range, indicating that there are significant differences between companies, at the individual and national levels, in terms of their CSR strategies [68]. The Refinitiv database will expand in the following years, thus increasing the sample size and the power of statistical procedures. International corporate governance is a promising area of research with the help of sophisticated and granular data.

The implications for regulators are significant. On the matter of board gender diversity, the European Parliament has formally adopted the new EU law on gender balance on

corporate boards [69], mandating 40% non-executive female board members by 2026. Norway, Spain, Finland, Quebec (Canada), Israel, France, Italy, and Belgium [70] already have gender board quotas between 33% and 50%. It is expected that other governments of G20 countries, outside of the European Union, will also consider this regulatory development. In this context, the national regulatory environment (civil law, common law, self-regulation, Islamic law, or any combination of these) could be a barrier or a facilitator of adopting gender equality legislation [71]. This is a promising avenue for future research.

The implications for companies are also significant. Board independence, which is defined in national regulation in numerous countries, can be mandated in specific proportions by national corporate governance codes (such as in the UK [72]). Board independence is associated with a set of ethical virtues that reinforce each other and support the final goal of the organization [73]. However, the effect of board independence on agency costs is contingent on factors such as strong CEO organizational identification [74]. This result has been confirmed in France (part of the G20), where the costs of board independence outweigh the benefits of having a higher proportion of independent directors [75]. In conclusion, board independence is not a universal solution to (potential) agency conflicts, so that national regulators can opt for a flexible approach on this matter.

Comparative research on corporate governance is tempted to assume that there is an "optimal" board structure [40] or an ideal system of corporate governance. This is not supported by scientific evidence. Some attributes that are considered "best practice" are costly to implement. Others are not culturally accepted in some countries, even if they are assumed to increase financial performance. This area of research is in continuous development and the present study enriches the quantitative evidence on the actual implementation of corporate governance systems in the most developed economies around the world.

**Funding:** This research received no external funding.

**Institutional Review Board Statement:** Not applicable.

**Informed Consent Statement:** Not applicable.

**Data Availability Statement:** The data used in this study are proprietary to Refinitiv and were accessed through an institutional subscription. The author does not have permission to share the data. The analytical algorithms were devised by the author of this paper.

**Conflicts of Interest:** The author declares no conflict of interest.

## Appendix A

A fragment of R code used to estimate between-country differences. *GovScore* is the Corporate Governance Pillar Score, *ISO* is the country abbreviation (categorical variable), *lnTA* is the natural logarithm of total assets converted to USD.

---
**Algorithm A1:** Code selection in RStudio for the ANCOVA and post-hoc analysis

---

```
library(rstatix) # version 0.7.0
anova_test(govscore_change1, GovScore ~ ISO + lnTA, type = 3, detailed = TRUE,
white.adjust = TRUE)

adj_means <- emmeans_test(govscore_change1, GovScore ~ ISO, covariate = lnTA)
get_emmeans(adj_means)

posthoc <- emmeans_test(govscore_change1, GovScore ~ ISO, covariate = lnTA,
p.adjust.method = "bonferroni")
```

---

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
