# Peer review of "Comparative Evidence on Corporate Governance Outcomes in the G20 Countries"

_world, doi:10.3390/world3040056_

Round 1

Reviewer 1 Report

Dear Author, 

Q1. Is it possible to detail your suggestion and provide a guide to the authors?

A1. Yes, the methodology was inferior, in my opinion. ANCOVA is not the most appropriate method. It could use multiple linear regression. *For your information ANCOVA and multiple linear regression are similar, but regression is more appropriate when emphasizing the dependent outcome variable. At the same time, ANCOVA is more appropriate when the focus is on comparing the groups from one of the independent variables But from my side, it is ok to overlook and proceed with its analysis as it stands. Finally, to present the findings, I would suggest a table so that the results are more readable.

Q2. What is the main question addressed by the research? Is it relevant and interesting?

A2. In lines 32-33, the author says: “The research question of this paper is: How different are several developed countries in their corporate governance outcomes, as of the year 2021? I think it is relevant and interesting.

Q3. How original is the topic? What does it add to the subject area compared with other published material?

A3. I’ve already mentioned that the topic is original.

Q4. Is the paper well written? Is the text clear and easy to read?

A4. The paper is clear and well written but needs grammar spell check.

Q5. Are the conclusions consistent with the evidence and arguments presented? Do they address the main question posed?

A5. Yes, the main question was posed and addressed clearly.

Here are some changes for improving your article and then proceeding with publication.

1. You must remove self-citations 21, 28, and 29.

2. Line 86 "Country rankings are presented according to several criteria." Name those criteria.

3. I suggest you improve the methodology and the presentation of the findings.

In general, the article is good enough. Make the changes I have pointed out to you.

Author Response

Dear Reviewer, 

I am grateful for the kind and relevant comments on my paper. By following your recommendations, the paper has improved!

Regarding the statistical methodology, I have written a paragraph in section 2.4 to explain why ANCOVA is adequate for the research question of this paper. While I totally agree that it is an inferior method to linear regression, ANCOVA is specific to group comparison, which is the main objective of my paper.

A careful spell check has been performed on the entire paper. The software Writefull (with a premium license) has been used by the author.

I have removed the self-citations.

The Results have been improved with new literature and commentaries. 

A table summarizing the Results has been added to section 4. Discussion.

The Conclusions have been significantly enhanced with implications for regulators and companies. For this purpose, eight articles have been added to the references, thus improving the bibliography.

Thank you again for your support!

Reviewer 2 Report

Very well study with good design and evidence support. The observation and insights from the data analysis is very helpful for more coming study on corporate governance issues if could be more deep discuss for the conclusion part. Further suggestion will be given about the managerial suggestion or policy suggestions. 

Author Response

(The authors gave the same response as above.)

Round 2

Reviewer 1 Report

Dear Author,

Congratulations on the improvement of your paper.

I'm happy that I helped you. It is a good paper.

Well done!!!